# Herbivore and pollinator body size effects on strawberry fruit quality

**Annika Leigh Salzberg**[1]*, **Heather Grab**[2], **Casey Hale**[1], **Katja Poveda**[1]

**1** Department of Entomology, Cornell University, Ithaca, New York, United States of America, **2** School of Integrative Plant Science, Cornell University, Ithaca, New York, United States of America

* as3565@cornell.edu

## Abstract

Land use change affects both pollinator and herbivore populations with consequences for crop production. Recent evidence also shows that land use change affects insect traits, with intraspecific body size of pollinators changing across landscape gradients. However, the consequences on crop production of trait changes in different plant interactors have not been well-studied. We hypothesized that changes in body size of key species can be enough to affect crop productivity, and therefore looked at how the field-realistic variation in body size of both an important pollinator, *Bombus impatiens* (Cresson), and a key pest herbivore, *Lygus lineolaris* (Palisot), can affect fruit size and damage in strawberry. First, we determined if pests vary in body size along land use gradients as prior studies have documented for pollinators; and second, we tested under controlled conditions how the individual and combined changes in size of an important pollinator and a key herbivore pest affect strawberry fruit production. The key herbivore pest was smaller in landscapes with more natural and semi-natural habitat, confirming that herbivore functional traits can vary along a land use gradient. Additionally, herbivore size, and not pollinator size, marginally affected fruit production—with plants exposed to larger pests producing smaller fruits. Our findings suggest that land use changes at the landscape level affect crop production not just through changes in the species diversity of insect communities that interact with the plant, but also through changes in body size traits.

## Introduction

Agriculture is the largest land use type in the world, comprising 40% of the earth's arable land [1]. This percentage continues to rise as natural lands are cleared for use as cropland or pasture [2]. Declines in natural lands threaten biodiversity and the essential ecosystem services wildlife within those habitats provide. Maintaining natural lands in the areas surrounding farmland provides critical services, such as pollination or pest suppression [3, 4]. Exploring the effects of landscape simplification on insects that provide crucial ecosystem services and dis-services is essential to the protection of our global food supply.

Land use change has a variety of effects, such as decreasing species richness and abundance in more simplified landscapes [3, 5–7]. A key effect land use change has on arthropods is its

**Data Availability Statement:** All data and code files are available from the Dryad database (DOI: 10.5061/dryad.4mw6m90j1).

**Funding:** This work was supported by the National Institute of Food and Agriculture, United States Department of Agriculture (https://www.nifa.usda.gov/) in the form of a grant [2019-67013-29367] to KP.

**Competing interests:** The authors have declared that no competing interests exist.

influence on insect traits not just at the community level [8], but also at the population level. Studies have shown a variety of effects of land use change on intraspecific body size. In pollinators such as bees, body size often decreases with increased landscape simplification [9, 10]. This effect can be driven by a low availability of resources that results in smaller pollinators, or due to an advantage in being smaller in lower-quality landscapes with fewer resources [9–12]. Pollinators with larger intraspecific body sizes have been shown to provide better pollination services, with larger bees able to better pollinate plants thereby increasing production of pollination-dependent crops, through increased yield or larger fruits [13].

Fluctuating body size of herbivores, on the other hand, has the potential to also change the effect they have on crops. Smaller herbivores potentially cause less damage to crops than larger herbivores. However, body size of herbivores has not been studied in the context of land use change, thus it is unknown whether landscape composition affects them, and subsequently whether this affects crop damage. Gaston & Lawton [14] found that body size of small insect herbivores can vary more widely than large herbivore species in the same ecosystem, and that their populations are more variable. Additionally, smaller insect herbivores may be less able to move across landscapes with fewer resources [15]. Studies have found a variety of responses of insect herbivores to changing landscapes in terms of abundance and species richness [16], though none have focused on body size in particular. Body size trends for pollinators, and the yet unknown body size trends for herbivores, can both have significant implications for the ecosystem services these insects provide. Keeping natural areas preserved around a farm to increase landscape complexity has great potential to affect body size of multiple insect guilds. Exploring the effects of those potential body size changes is crucial, as they have the potential for both positive and negative contributions to ecosystem services and yields.

To explore these questions, we chose strawberry as a study case. Strawberries and their interacting pollinators and herbivores are an ideal system to study the effects of landscape simplification on ecosystem services, mediated by changes in the functional traits of herbivores and pollinators. Strawberries are a high-value crop generally grown with high levels of inputs and are heavily managed [17]. Any amount of damage to fruits can render them unmarketable —and fruits can undergo damage in numerous ways. Two of the most important insect-related damages are inadequate pollination by bees (which results in seed underdevelopment) and the characteristic "cat-facing" damage caused when fed on by the tarnished plant bug, *Lygus lineolaris*. Tarnished plant bugs are a generalist herbivorous pest that cause damage to a wide range of crops–in strawberry specifically, they feed on developing fruits (just as the petals have fallen off the flower) which impedes proper development of those fed-upon achenes [18].

In strawberries, fruit weight can provide an accurate estimate of pollination rate and herbivore feeding, as strawberries are an aggregate accessory fruit comprising as many as 300 individual achenes [19]. Each achene must be fertilized and remain undamaged in order for the surrounding tissue to develop without major malformations that reduce overall yield and marketability [20]. Hence, the weight of a fruit is highly correlated with the number of pollinated, undamaged achenes [21]. An average of four pollinator visits per flower is required to achieve full pollination and maximum fruit weight [22], but densities as low as a single plant bug nymph per flower cluster can reduce fruit weight by 20% [20]. This system provides the perfect opportunity to examine the possible interactive effects on strawberry production by two key insect interactors.

Here, we examine the interactive effects of a key strawberry pest, the tarnished plant bug, and an important pollinator, the common eastern bumble bee, on fruit size. We expected to find similar results for our pollinator species as prior work—that larger pollinators would provide greater pollination services. For our pest species, we predicted that larger individuals would have greater capacity to cause damage. As both pollinators and herbivores can affect

development—and thus the marketability—of many crops, finding ways to mitigate damage caused by both poor pollination and pest feeding is particularly important. Specifically, we sought to answer four questions: first, does surrounding land use impact intraspecific traits like body size in agriculturally important herbivores? Second, does this size variation result in differential fruit damage—in other words, do larger herbivores cause more damage? Third, does pollinator body size have an effect on fruit size? And finally, do pollinator and herbivore body size variation have a combined or synergistic effect on fruit size?

## Methods

### Tarnished plant bug size across the landscape gradient

In summer 2019, 10 small farms growing strawberries as one of their crops in the Finger Lakes region of New York State (within a 50-mile radius of 42.697047, -76.712271) were selected across a landscape gradient. The landscape surrounding these sites consisted of 15% to 42% natural area at a 750m radius. Each site was visited once between August 12–30, and three of those sites were also visited once on July 2–3, though as tarnished plant bug populations were not large enough to enable enough specimens to be caught at that time, most individuals were caught in mid-late August. At each site, tarnished plant bugs were sweep-net collected both from within strawberry fields and in the weedy field margins around the strawberry crop. Landscape composition within a 500, 750, and 1000 meter radius around each collection site was determined via the 2019 USDA NASS CropScape Cropland Data Layer (nassgeodata.gmu. edu/CropScape). Multiple scales were chosen in order to determine which was most predictive of tarnished plant bug size, and these particular scales were chosen based on the most indicative scales found in a prior landscape study of this species [18]. The Cropland Data Layer provides crop-level land cover, which we then aggregated into broader land cover categories. Those land cover categories were agricultural, agricultural with pasture, natural forested, natural open, pasture, and urban (urban being defined as buildings and other developed areas like roads) cover. Strawberry field cover was included in the "agricultural" category, but was not analyzed separately because it is not an industry large enough in this area for it to be possible to analyze proportion strawberry in the landscape. A Mantel test revealed no spatial autocorrelation of sampling sites (Mantel $R$ = -0.1275, $P$ = 0.689, n = 10) Each tarnished plant bug's body size was measured, using pronotum width at its widest point as a proxy for whole body size [23]. Size was measured using an Olympus SZX10 stereo microscope and DP22 microscope digital camera, with Olympus CELLSENS Standard v.1.16 software used to measure bugs to the nearest 10 μm.

### Effect of bee size and tarnished plant bug size on strawberry fruit size

A greenhouse bioassay was conducted in spring 2021 to examine the effects of bee and tarnished plant bug size on strawberry pollination and damage, respectively. A total of 300 strawberry plants (*Fragaria x ananassa*, variety: Seascape, bare root plants sourced from Nourse Farms) were grown in the greenhouse (14 hours daylight, 70–74℉ day/70-66℉ night) until they flowered.

Three managed *Bombus impatiens* bumble bee colonies were sourced from Biobest (Biobest Standard hives). From these three hives, we constructed ten microcolonies (Fig 1) of 8 female workers each, all from the same source colony. Five microcolonies contained individuals with the smallest body sizes, selected by eye, and five contained individuals with the largest body sizes. Microcolonies were allowed to acclimate for one week with supplemental pollen and 30% sugar solution. Post-experiment, bee body size was measured to confirm that large and small colonies were significantly different in size by measuring intertegular distance (ITD), a

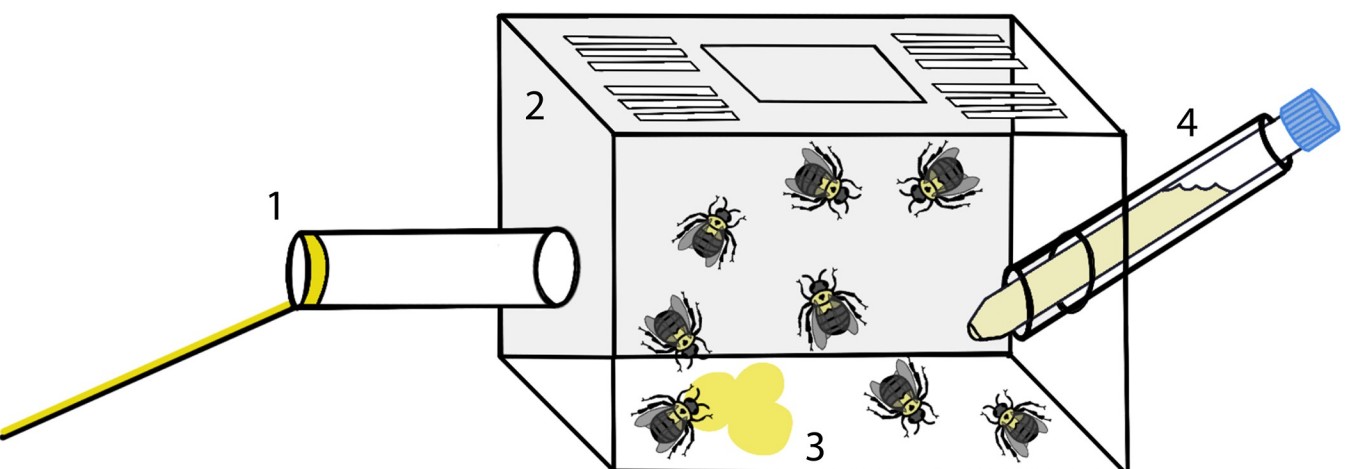

**Fig 1. Diagram of *B. impatiens* microcolony setup.** Consists of (1) Entrance/exit tube with ramp, with yellow tape placed around entry and on ramp to guide bees in; (2) Main chamber, plastic with lid on top and perforations for airflow (not pictured), with 8 individual bees; (3) Pollen balls (made in-lab with Arizona sourced pollen granules (CC Pollen High Desert brand) and 30% sucrose solution); (4) Tube with 30% sucrose solution, with small hole at base.

standard measure of body size in bees [24], Welch two-sample t-test comparing mean sizes of large versus small colonies, $t = 6.4746$, $df = 4.7168$, $P = 0.001628$). To ensure these colonies represented a standard range of bee sizes for the region, we compared them with other commercial hives used in other regional studies. In total, our average bee ITD was $3.47 \pm 0.36$mm (mean $\pm$ 1SD). Miller et al. [25] used commercial hives in the same region of New York State and found an average bee ITD of $3.34 \pm 0.45$mm (mean $\pm$ 1SD) and in nearby Southern Quebec, Gervais et al. [26] found an average ITD of $3.943 \pm 0.318$mm (mean $\pm$ 1SD). Both studies measured body size later in the season, to ensure that body size (a plastic trait affected by available resources) was a reflection of the landscape's resources and not the method by which they were commercially raised. Thus, these were used as a suitable proxy for regional bee size for this species and confirmed that our colonies fell within an average range.

All microcolonies were 'trained' on strawberry plants for a minimum of three days in order to ensure each individual had experience visiting strawberry flowers and had a built-up pollen load. Training consisted of colonies being kept open in a 24x36" mesh cage, with three or four blooming strawberry plants. These plants were individually switched out if no blooms were present at any point, and none of the training plants were used in the following trials. All colonies were kept in a growth chamber set at 14 hours daylight and 70–74˚F day temperature, 66–70˚F night temperature.

Pollination and herbivory trials were conducted within mesh cages placed outdoors and in the greenhouse, respectively (Fig 2). Pollination cages (24x36" white mesh pop-ups) were set up outdoors in the morning starting at around 9am each day–these trials were run outside in order to provide the most natural environmental conditions possible for the bees (wind, direct sun, etc.). Only plants with at least two open secondary flowers were used. Experimental plants were brought from the greenhouse and one or two were carefully inserted into each cage containing a microcolony. These plants were then observed until the first visit by a bee to one of the exposed secondary flowers. The plant was then removed from the cage, the pollinated flower was marked with yarn, and the other unpollinated secondary flower (at approximately the same stage of development) was marked as unpollinated. Pollination trials were run until around 12 pm each day, after which time visitation activity was negligible. Pollination trials were run for four days, until at least 60 pollinated plants per pollinator size group were

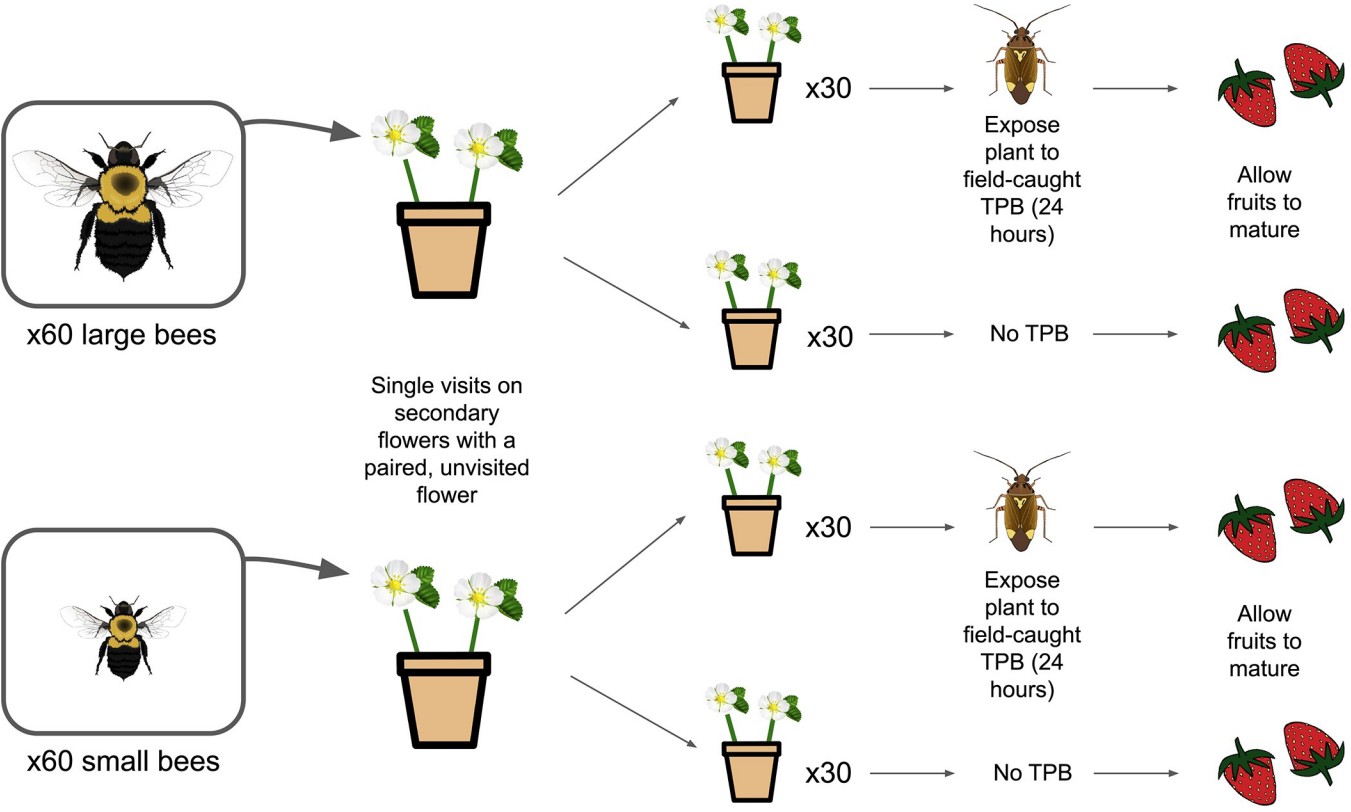

**Fig 2. Flowchart of bumble bee pollination and tarnished plant bug damage trials.**

acquired. Once plants were pollinated, they were transported back to the greenhouse and allocated into their plant bug trial group.

To determine the effect of *L. lineolaris* size on fruit damage and size, we conducted an experiment exposing developing fruits to individual bugs across a range of body sizes. These trials were conducted in the greenhouse. Approximately half of plants from each pollination size group, 30 plants pollinated by large bees and 34 plants pollinated by small bees, were placed in empty cages (no damage treatment). Each of the remaining pollinated plants, 30 pollinated by large bees and 30 pollinated by small bees, were placed in individual 12x12 cages along with a single wild-caught tarnished plant bug individual. The bugs for this experiment were collected from a subset of four sites used for the landscape trials, and a simple linear model was run to show that size did not differ between these sites ($F_{3,50} = 1.841$, $P = 0.1518$), eliminating the possibility of confounding effects from populations. These individuals varied in size from 1.930 to 2.328mm (pronotal width) and were randomly selected. This range is similar to the range seen across local landscapes—in 2019, field collections of tarnished plant bugs made across 10 sites of varying landscape complexity showed a range in size from 1.67 to 2.27mm (pronotal width). Due to plant bug mortality during trials, the final count of usable plants was 22 plants pollinated by large bees and 29 plants pollinated by small bees. As we were unable to determine exactly when the bugs in the cages had died, we were thus unable to say with certainty that the plant had been exposed for the full 24 hours–hence why these plants were removed from the experiment. The bugs were left to feed on the plants for 24 hours, at which point they were removed, frozen, and their pronotal width measured to the nearest 10 μm using an Olympus SZX10 stereo microscope and DP22 microscope digital camera, with

Olympus CELLSENS Standard v.1.16 software. Plants were then removed from small cages and placed together in large cages. This was done to enable greenhouse staff to water the plants, which were inaccessible in the small cages, and to eliminate the risk of stray pollinators in the greenhouse accessing them. Plants were watered regularly and monitored until fruits were fully developed. Once that stage was reached, each fruit was removed and weighed.

## Statistical analysis

### Tarnished plant bug size across the landscape gradient

We constructed linear mixed effects models to look at the effects of land cover on plant bug size, with pronotal width as the response variable and land cover at 750 meters and sex of the insect as predictors, with farm as a random effect. The sex of each plant bug was determined via presence or absence of an ovipositor. Models were constructed for all land cover types over all three landscape scales (500, 750, and 1000m), with sex as an additive factor and farm as a random factor. We only report effects on significant land cover variables. To identify potential correlations between the two significant land cover types, a Pearson's correlation test was run to determine correlation between urban and forest land cover.

### Effect of bee presence/absence and tarnished plant bug presence/absence on strawberry fruit size

To determine whether the presence or absence of pollinators and plant bugs affected final strawberry fruit weight, we constructed a linear mixed effects model with fruit weight as the response variable, and pest presence/absence and pollination treatment (visited or unvisited) as the two predictors. Microcolony number (which microcolony the bee came from) was used as a random factor.

### Effect of bee size and tarnished plant bug size on strawberry fruit size

To investigate how tarnished plant bug size and pollinator size affect strawberry fruit weight we constructed a linear mixed effects model with fruit weight as the response variable, and bug width and pollinator size treatment as the two predictors. Microcolony number (which microcolony the bee came from) was used as a random factor. This analysis was run on a subset of the data that only included fruits that were both exposed to a plant bug and had been pollinated. Fruit weight was used as a metric for the amount of damage done by the pest. We were unable to use surface damage as a metric because the damage caused by this pest looks almost identical to that caused by poor pollination, and we were unable to differentiate whether damage was from the plant bug pest or from improper pollination. Analyses were conducted in RStudio 2021.09.0 [27], using packages 'nlme' [28], and 'stats' [29]. Graphs were constructed in RStudio 2021.09.0 [27], using packages 'ggplot2' [30] and 'cowplot' [31].

## Results

### Landscape simplification increases tarnished plant bug size

For tarnished plant bugs collected across a landscape gradient, we found that both males and females were larger in landscapes with higher urban land cover ($F_{1,8} = 6.552$, $P = 0.0387$, Fig 3A, S1 Table). We also found that tarnished plant bugs from landscapes with higher forested land cover were marginally, though non-significantly, smaller ($F_{1,8} = 3.718$, $P = 0.0628$, Fig 3B, S2 Table). These two land cover types were negatively correlated with each other (Pearson's correlation, $t = -12.416$, df = 65, $P < 0.001$, correlation coefficient = -0.84), thus it is impossible

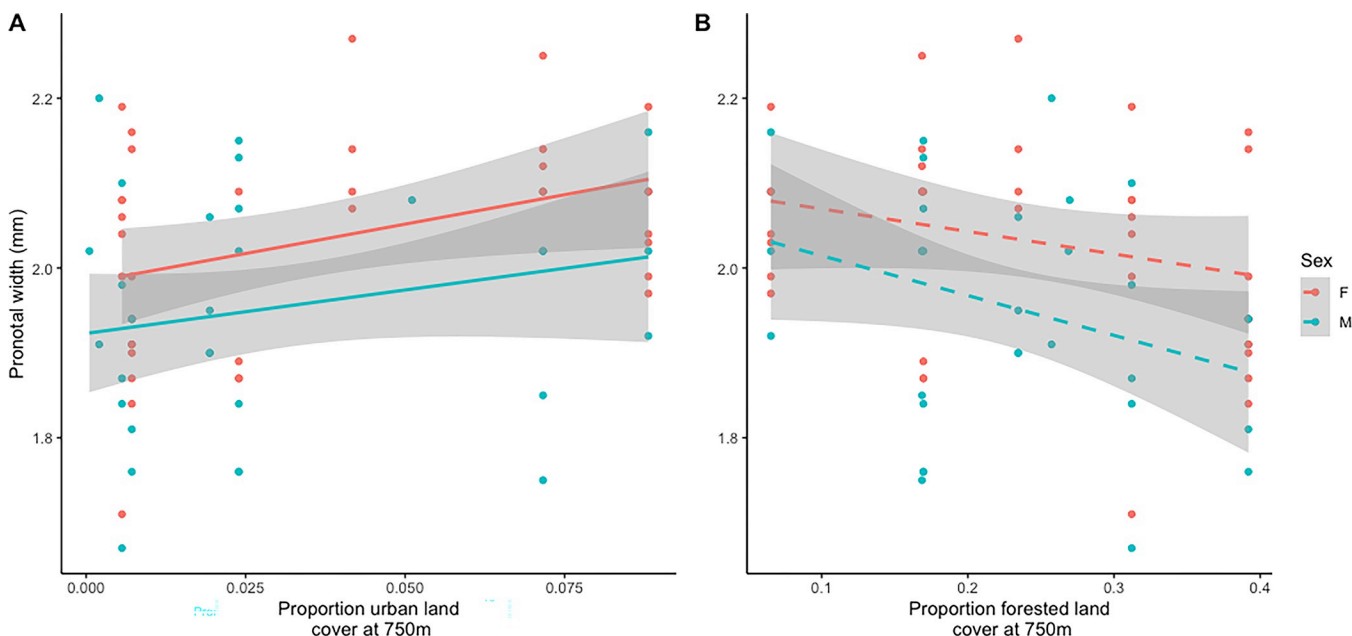

**Fig 3. Tarnished plant bug pronotal width at widest point across a landscape gradient.** Red points/line are female bugs, blue are males. **(A)** Urban land cover, trendlines = linear regressions, shaded area = 95% confidence area. **(B)** Forested land cover, trendlines = linear regressions, shaded area = 95% confidence area.

to tease apart if the effect is caused by one or the other. Female bugs were larger than males (Fig 3A: $F_{1,56}$ = 5.294, $P$ = 0.0251; Fig 3B: $F_{1,56}$ = 6.760, $P$ = 0.0119), but their relationship to land cover was the same (urban landcover interaction with sex: $F_{1,55}$ = 0.132, $P$ = 0.7175; forest cover interaction with sex: $F_{1,55}$ = 0.524, $P$ = 0.4722).

### Bee presence, but not tarnished plant bug presence, increases strawberry fruit size

Fruit weight of pollinated flowers was significantly higher than unpollinated flowers ($F_{1,198}$ = 38.046, $P$ < 0.001, Fig 4A). For all plants (both pollinated and unpollinated) plant bug presence did not affect fruit weight ($F_{1,198}$ = 0.0255, $p$ = 0.873, Fig 4B, S3 Table).

### Tarnished plant bug size, and not pollinator size, reduces strawberry fruit size

We found that there was no difference in the weight of fruits pollinated by large and small bees ($F_{1,40}$ = 0.2424, $P$ = 0.6252, Fig 5A). Because we found that pest size varies across the landscape gradient, we wanted to know if pest size affected the amount of damage they inflict on strawberries. We found that among pollinated fruits, exposure to larger tarnished plant bugs caused fruits to be marginally smaller ($F_{1,40}$ = 3.57, $P$ = 0.0661, Fig 5B).

Interestingly, there was no interaction between herbivore size and bee size ($F_{1,39}$ = 0.7082, $P$ = 0.4052, Figure A in S1 Fig), indicating that the effect of tarnished plant bug size on resulting fruits did not vary depending on the size of the pollinator.

## Discussion

This study sought to untangle the effects of land-use change on ecosystem services and disservices, as mediated by changes in intraspecific body size of an important pollinator and a key

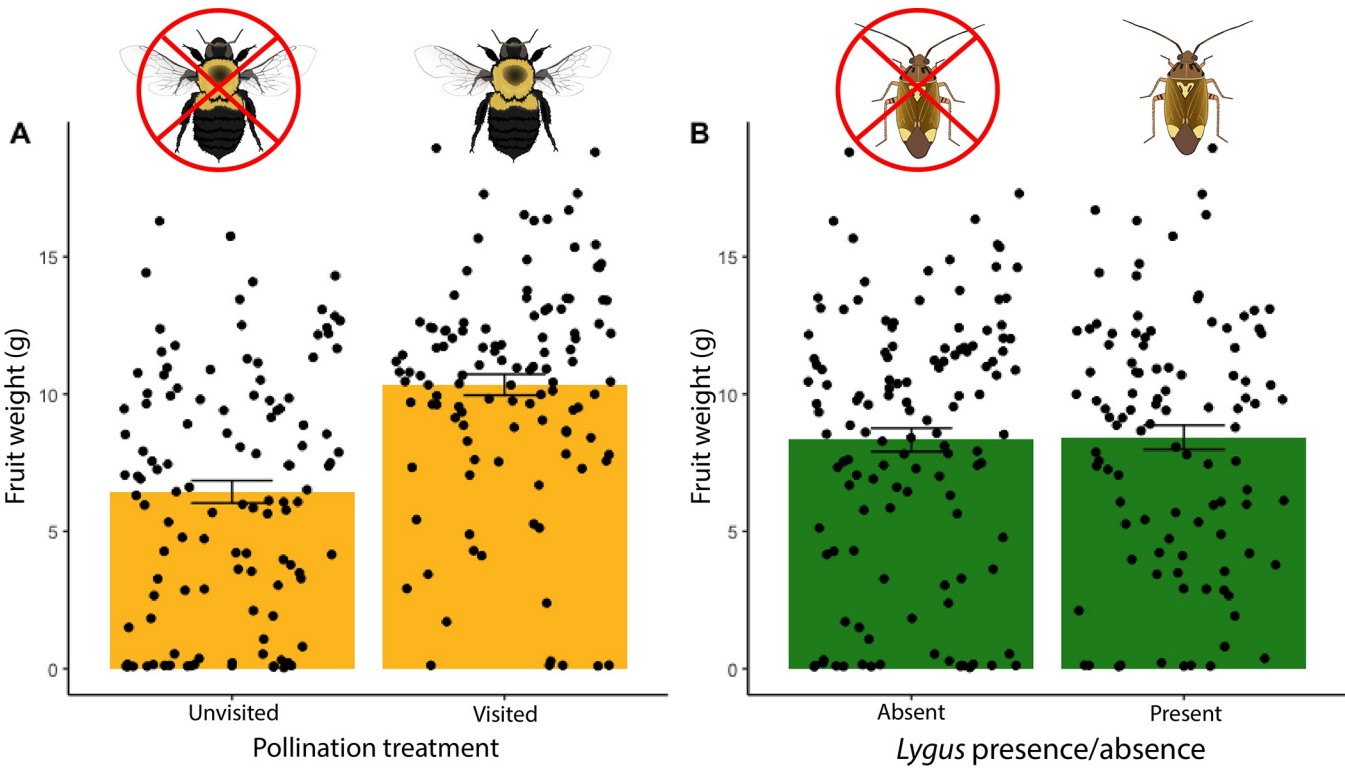

**Fig 4. (A)** Fruit weight in grams for flowers that were visited or not visited by a bumble bee (*** = p<0.001) **(B)** For all plants both pollinated and unpollinated, fruit weight in grams by presence or absence of tarnished plant bugs. Each plant was exposed to one individual for 24 hours. (p = 0.817).

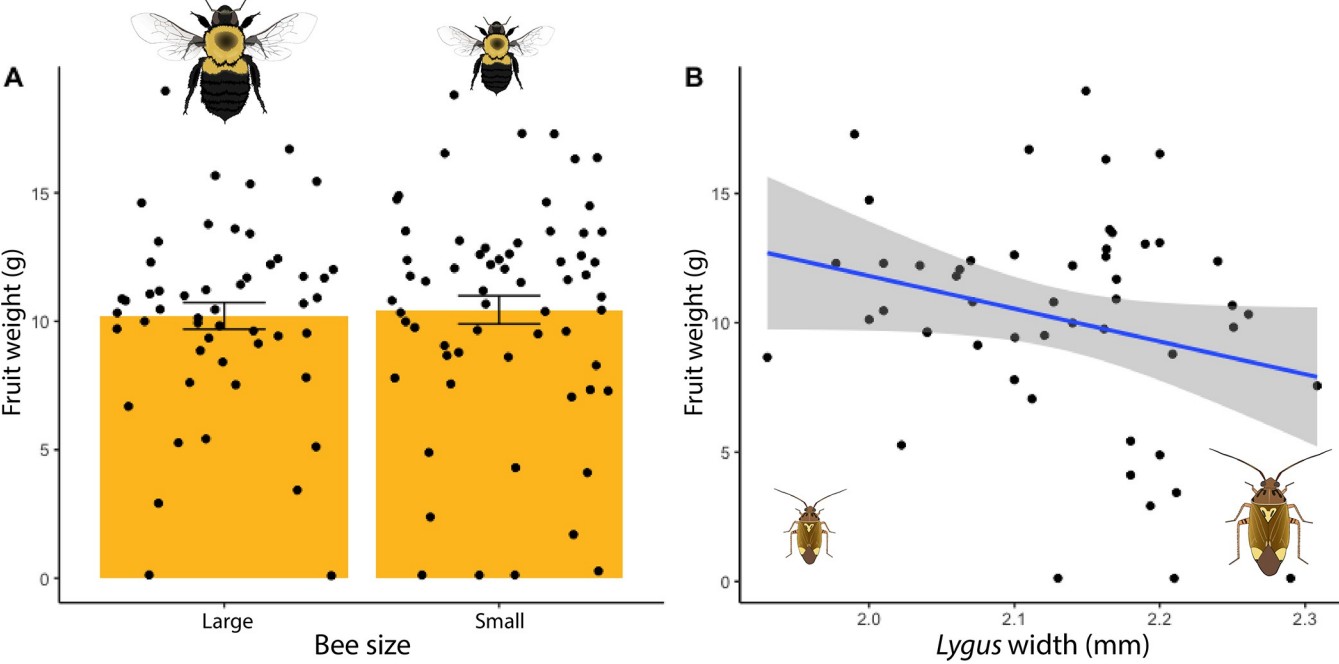

**Fig 5. (A)** Fruit weight by size of bumble bee visitor (p = 0.6252); **(B)** Fruit weight by size of tarnished plant bug the plant was exposed to for 24 hours (p = 0.0661). Both A and B only include data from fruits that were pollinated.

pest. We then asked whether those changes in ecosystem services and disservices provided by the two focal species would combine or synergize to create a greater positive or negative impact on fruit size.

First, we wanted to elucidate the effects of landscape composition on herbivore size. We found that tarnished plant bugs were smaller in landscapes with more natural forest and less urban area in the surrounding 750m around strawberry farms. This is an important finding, as there are virtually no studies looking at how intraspecific herbivore body size varies across a landscape gradient. Our results for this herbivore also contrast with studies done on pollinators, many of which have found increasing body sizes when landscape complexity increases [10, 32]. Tarnished plant bugs could be using urban areas, but since the gradient is so small (0.05–8.81%) this result is more likely a response to the much broader forest gradient (6.52–39.19%). There are many possible reasons for finding smaller tarnished plant bugs in more complex landscapes; some of these include potentially increased competition for resources [15], more natural enemies putting pressure on the pest population (though patterns vary [33, 34]), or fewer crop resources available for feeding. Additionally, tarnished plant bugs utilize perennial woody and forest edge habitat not only as an alternative area for suitable hosts but also for overwintering [35]. Larger individuals with greater dispersal capacity [36] may move more widely across the landscape than their smaller counterparts which may be constrained to areas near alternative hosts and overwintering sites.

Having confirmed that landscape composition affects pest size, we then wanted to know whether the size of the bug would differentially affect the growth of the developing fruit it was feeding on, resulting in smaller fruits. While prior studies have suggested a correlation between bee size and pollination efficiency [13, 37], we are unaware of any studies evaluating the same patterns in herbivores. We saw a trend that the presence of larger bugs resulted in smaller fruit regardless of pollination treatment. Fruit size went from ~13 grams when exposed to small bugs to ~7.5 grams when exposed to large bugs, a decrease of over 40%, though this effect was only marginally significant. These fruits were within the normal size range expected for this cultivar [38–40], with fruit size ranging from 0.04–18.96 grams. Even as a marginal effect, a 40% decrease in weight translates to important yield losses when scaled across an entire crop field. The effects we report are also likely to be conservative estimates, as our fruits were exposed to just a single bug over only 24 hours, while in the field, the fruit would be exposed for longer periods of time and potentially to more bugs.

We notably found no difference in fruit size between fruits exposed or not exposed to tarnished plant bugs. Though we find their presence/absence to result in the same range of fruit sizes, when the bugs *are* present, their size variation contributed to explaining the variation in fruit sizes observed with marginally smaller pollinated fruits produced in the presence of larger pests. Although it is unclear why we don't see a negative effect of overall pest presence, we see that there is a huge variation in fruit weight in our experiment, which could be caused by differences in pollination quality (due to limited pollen load on the bees from single visits) or fruit development. However, part of that variation seems to be explained by the variation in plant bug size. In field-realistic settings we may see a much wider range of fruit size changes— and it is unclear whether pollination and pest damage will scale up at the same rate suggesting an important direction for future research.

As many previous studies in different crops system have shown, bee size (both inter- and intra-specific) is frequently an important factor impacting fruit set and size [13, 41, 42]. Contrary to these prior studies, we found no difference in fruit weight between flowers pollinated by large versus small bees. This suggests that for this cultivar of strawberry and its interaction with this particular species of pollinator, *Bombus impatiens*, intraspecific body size of the pollinator does not matter. This result could also be due to the single visit nature of this study,

where pollen loads on the bees were likely to be lower than in other studies where bees were allowed unlimited visitation. Pollen load could also interact with other traits like hairiness, position of pollen loads, behaviors, and flower traits to influence pollination outcomes [43–46]. With further regards to pollen load, one might think that the lack of difference between large and small bees is a result of the lack of diversity in the source of the pollen, as it was previously shown that smaller bees carry higher diversity of pollen [26]. In natural systems, bees of different sizes may carry different pollen diversity, which could impact pollination through an increase of sigma blocking by heterospecific pollen transfer and dilution of conspecific pollen. However, in our system it is important to take into account that strawberries are autopollinated, and do not require cross pollination to set fruit. Thus, having access to only the strawberry clones in the cages would not be expected to have a major effect on fruit set unless insufficient pollen was transferred (which we have addressed above).

Our study additionally did not examine the effects of land-use change on *Bombus impatiens* body size. Prior studies have examined the effect of landscape on multiple species of wild bees, finding mixed results. Landscape effects on bee size seems to be very context-dependent, with some studies finding that bees are larger in complex landscapes [47, 48], and others finding they are smaller [49]. For bumble bees specifically, conflicting forces act on colonies resulting in mixed body size effects. There are benefits to colonies creating larger workers in more fragmented landscapes, as many larger bees can fly further and access a greater variety of resources–but simultaneously, colonies may not have the resources to generate larger workers in simpler, resource-poor landscapes, and thus create smaller workers [12, 50–52]. However, in our system (strawberry in New York State) we found that the variation in bee size found in our colonies has no effect on pollination efficacy. Therefore, regardless of bee size variation across a gradient of landscape complexity within our system, we do not expect to see any effect on strawberry fruit pollination. However, increased landscape complexity has been shown to have many other beneficial impacts on pollinator communities beyond body size, and can have beneficial effects beyond intraspecific body size such as increased species diversity [7, 53] and a greater diversity of foraging resources [37, 54].

Our results support the large body of literature showing the benefits of natural areas for agricultural production. We show that the presence of natural areas around a farm can lead to a decrease in the body size of herbivores with consequences for crop yield. Alternate methods of pest suppression, such as the maintenance of land cover diversity, have the potential to lessen the usage of pesticides and lower costs for growers. Previous work in the New York State region has shown how greater landscape complexity also has a wide variety of positive impacts on pollinators—including increased bee richness, abundance, larger bee body size, and increased pollination [32, 53]—as well as decreases in tarnished plant bug abundance [18]. The benefits to pollinators and the potential for decreased herbivory could provide double benefits to growers whose farms are situated in more complex landscapes. Strawberry growers should consider keeping some forested land around their fields, as prioritizing the maintenance of a complex landscape has the potential to provide benefits for both pollination services and pest suppression alike.

## Supporting information

**S1 Table. Results of linear mixed-effects model fit by REML comparing *Lygus* pronotal width and urban cover at 750 meters.** Fixed effects: *Lygus* size ~ Urban cover at 750m + Sex. (DOCX)

**S2 Table. Results of linear mixed-effects model fit by REML comparing *Lygus* pronotal width and natural forest cover at 750 meters.** Fixed effects: *Lygus* size ~ Natural cover at

750m + Sex.
(DOCX)

**S3 Table. Results of linear mixed-effects model fit by REML comparing strawberry fruit weight with pollination treatment and tarnished plant bug presence/absence.** Fixed effects: Strawberry fruit weight (g) ~ pollination treatment + *Lygus* presence/absence.
(DOCX)

**S1 Fig. Tarnished plant bug size by fruit weight and pollination.** Fruit weight by size of tarnished plant bug the plant was exposed to for 24 hours. Red points are those flowers visited by large bees (first panel), green points are those visited by small bees (second panel), and blue are unvisited flowers (third panel). Trendlines = linear regressions, shaded area = 95% confidence area.
(DOCX)

# Acknowledgments

Huge thanks to all the people who helped with bumble bee single visit trials—Hayley Schroeder, Dr. Jules Davis, and Dr. Laura Figueroa. Thanks as well to the Cornell Greenhouse staff for plant care and support.

# Author Contributions

**Conceptualization:** Heather Grab, Katja Poveda.

**Data curation:** Annika Leigh Salzberg, Casey Hale.

**Formal analysis:** Annika Leigh Salzberg.

**Funding acquisition:** Katja Poveda.

**Investigation:** Annika Leigh Salzberg.

**Methodology:** Annika Leigh Salzberg.

**Supervision:** Katja Poveda.

**Visualization:** Annika Leigh Salzberg.

**Writing – original draft:** Annika Leigh Salzberg.

**Writing – review & editing:** Annika Leigh Salzberg, Heather Grab, Casey Hale, Katja Poveda.

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
