## [Decision Letter · Decision Letter 0]

17 Jan 2024

PONE-D-23-40097The impact of land use change on intraspecific variation in pollinators and herbivores and its effect on strawberry productionPLOS ONE

Dear Dr. Salzberg,

Thank you for submitting your manuscript to PLOS ONE. After careful consideration, we feel that it has merit but does not fully meet PLOS ONE’s publication criteria as it currently stands. Therefore, we invite you to submit a revised version of the manuscript that addresses the points raised during the review process.

We look forward to receiving your revised manuscript.

Kind regards,

Ali Akbar Ghasemi-Soloklui, Ph.D

Academic Editor

PLOS ONE

Journal Requirements:

3. Please expand the acronym “NIFA” (as indicated in your financial disclosure) so that it states the name of your funders in full.

Additional Editor Comments:

Dear authors

Please perform the suggested revision and send it back as soon as possible.

Reviewers' comments:

Reviewer's Responses to Questions

**Comments to the Author**

1. Is the manuscript technically sound, and do the data support the conclusions?

Reviewer #1: Partly

Reviewer #2: Yes

2. Has the statistical analysis been performed appropriately and rigorously? 

Reviewer #1: Yes

Reviewer #2: Yes

3. Have the authors made all data underlying the findings in their manuscript fully available?

Reviewer #1: Yes

Reviewer #2: Yes

4. Is the manuscript presented in an intelligible fashion and written in standard English?

Reviewer #1: Yes

Reviewer #2: Yes

5. Review Comments to the Author

Reviewer #1: The manuscript “The impact of land use change on intraspecific variation in pollinators and herbivores and its effect on strawberry production” proposes to study the effect of the body size of a pollinator species and a pest species on strawberry production. Overall the manuscript is interesting, however, several major concerns need to be addressed.

### Major concerns ###

1) What is the distance between sites? Are they spatially independent? What about spatial autocorrelation?

2) I don't understand where the pests were collected in the study sites, in which crop, etc. Moreover, you only collected the pests, not the pollinators. Without this element, the field study is missing half the story. I.e., what was the effect of landscape gradient on bumble bee size?

3) You selected individuals with the smallest body size (selected by eye) vs. individuals with the largest body size (also selected by eye) to make up your experimental design. Do you know why your individuals are smaller? Are they of lower individual quality (in terms of physiology)? If so, is morphology (body size) the driver of pollination efficiency or physiology?

4) It is not clear if the small pests come from specific locations and the large pests come from other specific locations. If this is the case, you have a confounding effect with the population, and this could disturb the experimental design based on the body size alone.

5) In the cage pollination experiment, you put plants in cages with a colony of bumble bees, if I understand correctly. However, with this method, there is little pollen diversity carried by the bumble bees, since it comes from the same plant or another genetically similar plant. Maybe this helps explain why you find no difference in fruit weight between small and large bumble bees. Perhaps under field conditions, large bumble bees could carry larger quantities and more diverse pollen since they could potentially travel farther to acquire pollen than small individuals.

6) The discussion could be improved by adding a comparison of the results obtained with the literature on strawberry production. Are the results realistic in terms of fruit weight? Another point that is missing is fruit set. In pollination experiment, the effect on fruit set is usually tested first, and a stronger effect is found than on fruit quality (excepted for seed number). This should be discussed in the context of the present study.

### Point-by-point comments ###

Title: The title should clearly state that the study considered a single pollinator species and a single pest species. Also, "The impact of land use change on strawberry production" is not really tested in this study. I found the title unrelated to the work done.

Lines 63-65 : It is not clear if you are referring to species differences in body size, or a decrease in body size of individuals of a species.

Lines 83-93 : I found this part a bit repetitive.

Lines 98-124 : Most of the text presented here is not related to Materials and Methods, but should be moved to the Introduction as material justifying the use of strawberry as a "study case".

Lines 100-102: perhaps use hectares instead of acres?

Line 116: The reference is in the wrong format.

Line 115: I think a parenthesis is missing.

Line 125: What was the effect of landscape gradient on bumble bee size? Without this element, the field study is missing half the story.

Lines 126-127: You should include the latitude and longitude for transparency on the overall location of the study.

Line 131: Why are these radii chosen? It would be useful to determine this based on the foraging range of pollinators and pests, with references.

Line 133: I don't understand "Categories of land cover quantified were agricultural, ..."

Line 135: Again, it would be great to justify the choice of such landscape parameters based on assumptions of landscape effects and references.

Lines 135-137: I don't understand where the pests were collected, in what crop, etc. Lines 156-161: In my opinion, this analysis does not present an argument that the bees used in the study are representative of the standard range of bee sizes for the region. They compared them to commercial bees imported from other regions, not to the wild population of the region.

Line 176: The distinction between outdoor and greenhouse is not well shown in Figure 2.

Lines 179-182: It might be interesting to look at the visitation history of pollinators in the cage experiments if you collected this data. For example, if a bumble bee's first visit is to a secondary flower used in the experiment, the resulting fruit may be different than if the bumblebee had previously visited other flowers. I acknowledge that this may be complex and that the sample size may not allow for this type of analysis.

Lines 197-198: It is not clear if the small pests come from specific locations and the large pests come from other specific locations. If this is the case, you have a confounding effect with the population, and this could confound the experimental design based only on body size.

Line 199: Did you discard the plants on which the lygus died within 24 hours? This is unclear to me.

Line 203-205: Is there a species here? I don't understand this sentence.

Line 212: Please explain how you determined the sex of the individuals.

Lines 230-232: If I understand well, you have quite small sample sizes for fruits exposed to lygus and pollinated. Did you check for normality in these small sample sizes?

Line 242: It is not clear what you mean by developed land cover. It would be great to present what you are including as land use type.

Line 243: You use lower case for numbers here and upper case for numbers. Standardize.

Lines 244-245: You need to add that this result is not significant and dash the prediction lines in Figure 3B.

L250 : Please close the parenthesis.

L258-259 : You have to present the direction of the effect. Positive? Negative?

Reviewer #2: The growth of the body in insects such as the bumble bees is stopped due to the existence of the exoskeleton after the larval and pupal stage turn to adult, so feeding in the pre-adult stage is very important, which is often done by nurse bees. On the other hand, because the feeding of the larvae is very dependent on the activity of the nurse and forager bees, therefore, the quality of the plant host is not the only effective factor.

For the growth of pollinator bees, the ratio of different plant pollens and the presence of mineral elements and amino acids are also necessary. In summary, many factors are important in changing the body size of pollinator bees that were not considered in this experiment. However, The results are acceptable.

6. PLOS authors have the option to publish the peer review history of their article (what does this mean?). If published, this will include your full peer review and any attached files.

Reviewer #1: No

Reviewer #2: **Yes: **Dr. Alireza Monfared

---

## [Author Response · Author response to Decision Letter 0]

4 Mar 2024

Dear Dr. Ghasemi-Soloklui and PLOS ONE reviewers,

We would like to thank you all for your careful review and valuable comments on our manuscript! We feel that your guidance has helped significantly improve our work. Below, please find our responses to all points raised by reviewers.

Addressing comments from the editor:

 The files have been checked and renamed in accordance with PLOS ONE’s style requirements. Additional edits were also made, including adjusting font size/style of headings, properly formatting author affiliations, removing funding sources from acknowledgements section, and adding supplemental figure information at the end of the document. 

2. If you’ve not already done so, consider depositing your raw data in a repository to ensure your work is read, appreciated, and cited by the largest possible audience. 

 We have uploaded our data to Dryad, DOI: 10.5061/dryad.4mw6m90j1, and we will change the settings to make it publicly available if our manuscript is accepted.

3. Please expand the acronym “NIFA” (as indicated in your financial disclosure) so that it states the name of your funders in full.

 Thank you, we have made this change.

4. When completing the data availability statement of the submission form, you indicated that you will make your data available on acceptance. We strongly recommend all authors decide on a data sharing plan before acceptance, as the process can be lengthy and hold up publication timelines. Please note that, though access restrictions are acceptable now, your entire data will need to be made freely accessible if your manuscript is accepted for publication. 

 We have uploaded all our data and code to Dryad, DOI: 10.5061/dryad.4mw6m90j1, and we will make all data publicly available if our manuscript is accepted. Temporary access for editors: https://datadryad.org/stash/share/7Uje8Cu-TV_gPw6VCTIaNNZCP7yFt1uwvYm7EnORr14

The supporting information files have been updated to conform to PLOS ONE’s naming guidelines.

Addressing Additional Editor Comments:

Reviewer #1: 

### Major concerns ###

1) What is the distance between sites? Are they spatially independent? What about spatial autocorrelation? 

Our sites were spatially independent, and I have added the results of a Mantel test for spatial autocorrelation into the methods section to show this. Those numbers can be found in the “Tarnished plant bug size across the landscape gradient” section. Thank you!

2) a)I don't understand where the pests were collected in the study sites, in which crop, etc. b) Moreover, you only collected the pests, not the pollinators. Without this element, the field study is missing half the story. I.e., what was the effect of landscape gradient on bumble bee size?

a) Thank you for pointing out this oversight! I have clarified in the text where the pest were collected, and that they were collected in strawberry fields and the weedy field margins around them. 

b) To address your second point – no, we did not collect pollinators, so unfortunately we do not have access to body size data on worker Bombus impatiens individuals. However, work has been done on multiple Bombus species (and many other species of wild bees) examining their body size across landscape, and I have added references in the discussion section to multiple papers detailing that work. 

3) You selected individuals with the smallest body size (selected by eye) vs. individuals with the largest body size (also selected by eye) to make up your experimental design. Do you know why your individuals are smaller? Are they of lower individual quality (in terms of physiology)? If so, is morphology (body size) the driver of pollination efficiency or physiology?

This is an excellent question, but unfortunately beyond the scope of this paper to answer based on the experiments we ran. To postulate a possible answer however - I would not expect smaller individuals to necessarily be of lower individual quality, as there can be both benefits and detriments to being larger or smaller. This may be more of a concern if we were to test pollination efficiency in the field (as larger bees can often fly further, perhaps accessing more pollen resources and making pollination efficiency more of a physiology question). However, since there was no impediment to accessing the flowers based on distance to be flown (the microcolonies were about 1-2 feet away from the test plants) we can say with reasonable certainty that morphology rather than physiology was the force at play in pollination efficiency in our study.

4) It is not clear if the small pests come from specific locations and the large pests come from other specific locations. If this is the case, you have a confounding effect with the population, and this could disturb the experimental design based on the body size alone.

I believe you’re discussing the bugs used in our second experiment, the pollination x pest damage experiment. I have added text in the methods explaining where the bugs for this experiment were captured, and I have added an additional analysis showing that for the pests collected for this bioassay, insect size did not differ by site. Thank you for pointing out this omission!

5) In the cage pollination experiment, you put plants in cages with a colony of bumble bees, if I understand correctly. However, with this method, there is little pollen diversity carried by the bumble bees, since it comes from the same plant or another genetically similar plant. Maybe this helps explain why you find no difference in fruit weight between small and large bumble bees. Perhaps under field conditions, large bumble bees could carry larger quantities and more diverse pollen since they could potentially travel farther to acquire pollen than small individuals.

 Thank you for this comment! It’s true that the bees were exposed to less pollen diversity than they would be in field conditions. However, while some plant species require cross pollination to set fruit, strawberry does not (as an autopollinating plant). So, having access to only the strawberry clones in the cages would not be expected to have a major effect on fruit set unless insufficient pollen was transferred (which is possible). I have added text to the discussion addressing this point, thank you!

6) The discussion could be improved by adding a comparison of the results obtained with the literature on strawberry production. Are the results realistic in terms of fruit weight? 

Another point that is missing is fruit set. In pollination experiment, the effect on fruit set is usually tested first, and a stronger effect is found than on fruit quality (excepted for seed number). This should be discussed in the context of the present study.

Thank you for bringing up these points! I was able to find a few papers that reported individual berry weight for the variety used in our study (Seascape) and have added reference that our berries were similar in size to those in the studies I found, and thus realistic in terms of fruit weight. 

To address your second point, while we did not explicitly look at fruit set, we had fruits that were at nearly 0 weight – so in that sense, we are looking at the combined effect of fruit set and fruit weight together. However, I have also added information and citations into the discussion regarding other studies that have looked at fruit set as well. Thank you!

### Point-by-point comments ###

Title: The title should clearly state that the study considered a single pollinator species and a single pest species. Also, "The impact of land use change on strawberry production" is not really tested in this study. I found the title unrelated to the work done.

I have reworded the title to more accurately describe the work done, thank you!

Lines 63-65 : It is not clear if you are referring to species differences in body size, or a decrease in body size of individuals of a species.

This was in reference to intraspecific body size, and has been clarified in the text. Thank you!

Lines 83-93 : I found this part a bit repetitive.

With this final paragraph of the introduction, we sought to be very explicit in what our hypotheses were and what exact questions we sought to answer. Though this may be a bit repetitive of other points made in the intro, we think it is worth the extra time taken to be very clear about our aims. As the reviewer does not make a specific request as to what should be taken out, and I’m not comfortable removing anything in this paragraph as I feel it would take away from the clarity of our paper, I have not made any changes to this paragraph.

Lines 98-124 : Most of the text presented here is not related to Materials and Methods, but should be moved to the Introduction as material justifying the use of strawberry as a "study case".

Thank you for this suggestion! We had debated whether to put this section of text in the introduction or the methods, and based on your request we have moved it to the Introduction instead. 

Lines 100-102: perhaps use hectares instead of acres?

I have changed the measurements based on your suggestion, thank you.

Line 116: The reference is in the wrong format. 

Thank you, this has been corrected.

Line 115: I think a parenthesis is missing. 

I have replaced the commas with parentheses as suggested.

Line 125: What was the effect of landscape gradient on bumble bee size? Without this element, the field study is missing half the story.

^ I address this comment in the “major concerns” section above, see my response there.

Lines 126-127: You should include the latitude and longitude for transparency on the overall location of the study.

Thank you, I have added the latitude and longitude of the midpoint of all sites to clarify the location!

Line 131: Why are these radii chosen? It would be useful to determine this based on the foraging range of pollinators and pests, with references. Line 135: Again, it would be great to justify the choice of such landscape parameters based on assumptions of landscape effects and references.

Thank you for pointing out this oversight – I have added an explanation of why these radii were chosen and added the reference we used to make these choices.

Line 133: I don't understand "Categories of land cover quantified were agricultural, ..."

Thank you for pointing out this lack of clarity, I have elaborated on how these categories were aggregated from the more granular Cropland Data Layer.

Lines 135-137: I don't understand where the pests were collected, in what crop, etc. 

Thank you for pointing out these missing details – they have been added to the manuscript!

Lines 156-161: In my opinion, this analysis does not present an argument that the bees used in the study are representative of the standard range of bee sizes for the region. They compared them to commercial bees imported from other regions, not to the wild population of the region.

While it’s true that the studies referenced used commercially acquired hives, both studies took body size measurements later in the season. As Gervais et al. states, “Note that we restricted analyses regarding the morphology of workers to individuals captured at least four weeks after their colony was placed in the field to ensure that the potential effects on morphometrics would be associated to landscape variables, not the commercial rearing process.” While it would have been preferable to compare our bees to wild-caught individuals, data on wild Bombus impatiens body size has not been collected in regional populations, so unfortunately we are unable to make this comparison.

Line 176: The distinction between outdoor and greenhouse is not well shown in Figure 2.

Since these experiments were both done within cages, it didn’t seem necessary to detail in the figure that the cages were placed outside versus inside the greenhouse – instead, I have added more text to the methods section to clarify where exactly the cages were placed and explained why this was done. From other comments made by the reviewer I can see that I did not include enough detail to be clear about the methods of this part of the study, so hopefully my clarifications throughout the text resolve this issue to the satisfaction of the reviewer!

Lines 179-182: It might be interesting to look at the visitation history of pollinators in the cage experiments if you collected this data. For example, if a bumble bee's first visit is to a secondary flower used in the experiment, the resulting fruit may be different than if the bumblebee had previously visited other flowers. I acknowledge that this may be complex and that the sample size may not allow for this type of analysis.

Thank you for this comment! The bees involved in the pollination experiment were purposefully trained on strawberry plants in the greenhouse (within their microcolony cages), and while we cannot be 100% certain that each bee did in fact visit one of the training plants, from my anecdotal memory of the pollination trials I did see most microcolony bees had pollen loads from the test plants they’d built up during the training period. However, visit history and pollen load were not recorded and thus we are unable to examine this question.

Lines 197-198: It is not clear if the small pests come from specific locations and the large pests come from other specific locations. If this is the case, you have a confounding effect with the population, and this could confound the experimental design based only on body size.

This is a valid concern, and to remedy it I have added an additional analysis showing that for the pests collected for this bioassay, insect size did not differ by site. Thank you for pointing out this omission! 

Line 199: Did you discard the plants on which the lygus died within 24 hours? This is unclear to me.

Thank you, yes we did discard those plants from the analysis – I have added text to clarify this.

Line 203-205: Is there a species here? I don't understand this sentence.

Thank you for pointing this out, this was a grammatical structure error in the sentence! There is no missing species, just an oversight in my editing. I have changed the sentence to clarify that we placed all the plants together into larger cages in order to let the greenhouse staff keep the plants watered but to prevent any stray insects in the greenhouse from affecting the plants in any way.

Line 212: Please explain how you determined the sex of the individuals.

Thank you, this has been added.

Lines 230-232: If I understand well, you have quite small sample sizes for fruits exposed to lygus and pollinated. Did you check for normality in these small sample sizes?

Yes, we ran Shapiro-Wilk normality tests, which showed that even though these subsets of data were small, the residuals were normally distributed.

Line 242: It is not clear what you mean by developed land cover. It would be great to present what you are including as land use type.

I agree this was unclear – to remedy this, I added further clarification in the methods section when I initially described land cover categories. In line 242 as referenced in this comment, I changed the wording to align with the other mentions of urban cover to take out the word “developed.” I was using developed as a synonym, but as this was not consistent throughout the rest of the paper it added confusion and thus I removed it. Thank you! 

Line 243: You use lower case for numbers here and upper case for numbers. Standardize.

I have changed all figure references to be standardized as upper case, thank you for noticing this!

Lines 244-245: You need to add that this result is not significant and dash the prediction lines in Figure 3B.

Thank you, these changes have been made.

L250 : Please close the parenthesis.

Done, thank you!

L258-259 : You have to present the direction of the effect. Positive? Negative?

This

---

## [Decision Letter · Decision Letter 1]

7 May 2024

PONE-D-23-40097R1Larger pests but not pollinators impact strawberry fruit qualityPLOS ONE

Dear Dr. Salzberg,

Thank you for submitting your manuscript to PLOS ONE. After careful consideration, we feel that it has merit but does not fully meet PLOS ONE’s publication criteria as it currently stands. Therefore, we invite you to submit a revised version of the manuscript that addresses the points raised during the review process.

We look forward to receiving your revised manuscript.

Kind regards,

Ali Akbar Ghasemi-Soloklui, Ph.D

Academic Editor

PLOS ONE

Journal Requirements:

Reviewers' comments:

Reviewer's Responses to Questions

**Comments to the Author**

1. If the authors have adequately addressed your comments raised in a previous round of review and you feel that this manuscript is now acceptable for publication, you may indicate that here to bypass the “Comments to the Author” section, enter your conflict of interest statement in the “Confidential to Editor” section, and submit your "Accept" recommendation.

Reviewer #1: All comments have been addressed

2. Is the manuscript technically sound, and do the data support the conclusions?

Reviewer #1: Yes

3. Has the statistical analysis been performed appropriately and rigorously? 

Reviewer #1: Yes

4. Have the authors made all data underlying the findings in their manuscript fully available?

Reviewer #1: Yes

5. Is the manuscript presented in an intelligible fashion and written in standard English?

Reviewer #1: Yes

6. Review Comments to the Author

Reviewer #1: I have read this new version of the manuscript and the responses to the comments. I thank the authors for their great job addressing concerns. The manuscript is substantially improved and I believe it will make a great contribution to Plos One.

The only concern I have is the new version of the title. I think that the title "Larger pests but not pollinators impact strawberry fruit quality" is a bit misleading and gives the impression that animal pollination doesn’t matter for strawberry, which is contrary to the results of this study and the litterature.

7. PLOS authors have the option to publish the peer review history of their article (what does this mean?). If published, this will include your full peer review and any attached files.

Reviewer #1: No

---

## [Author Response · Author response to Decision Letter 1]

21 May 2024

Dear Dr. Ghasemi-Soloklui and PLOS ONE reviewers,

We would like to thank you all for your careful review and valuable comments on our manuscript! We feel that your guidance has helped significantly improve our work. Below, please find our responses to all points raised by reviewers.

Reviewer #1: I have read this new version of the manuscript and the responses to the comments. I thank the authors for their great job addressing concerns. The manuscript is substantially improved and I believe it will make a great contribution to Plos One.

The only concern I have is the new version of the title. I think that the title "Larger pests but not pollinators impact strawberry fruit quality" is a bit misleading and gives the impression that animal pollination doesn’t matter for strawberry, which is contrary to the results of this study and the literature.

Thank you for voicing your concern on the title – we agree, the grammar of the title was unclear, and we have modified the title to remove the ambiguity that might imply that animal pollination isn’t important for strawberry. We hope the new title, “Herbivore and pollinator body size effects on strawberry fruit quality” is a better representation of the paper and that this is acceptable to the reviewer. 

Thank you!

Best regards,

Annika Salzberg

---

## [Editor Report · Decision Letter 2]

30 May 2024

Herbivore and pollinator body size effects on strawberry fruit quality

PONE-D-23-40097R2

Dear Dr. Salzberg,

We’re pleased to inform you that your manuscript has been judged scientifically suitable for publication and will be formally accepted for publication once it meets all outstanding technical requirements.

Kind regards,

Ali Akbar Ghasemi-Soloklui, Ph.D

Academic Editor

PLOS ONE
---

## [Editor Report · Acceptance letter]

7 Jun 2024

PONE-D-23-40097R2 

PLOS ONE

Dear Dr. Salzberg, 

I'm pleased to inform you that your manuscript has been deemed suitable for publication in PLOS ONE. Congratulations! Your manuscript is now being handed over to our production team.

Kind regards, 

on behalf of

Dr. Ali Akbar Ghasemi-Soloklui 

Academic Editor

PLOS ONE